# B-Site-Ordered and Disordered Structures in A-Site-Ordered Quadruple Perovskites RMn_3_Ni_2_Mn_2_O_12_ with R = Nd, Sm, Gd, and Dy

**DOI:** 10.3390/molecules29235488

**Published:** 2024-11-21

**Authors:** Alexei A. Belik, Ran Liu, Masahiko Tanaka, Kazunari Yamaura

**Affiliations:** 1Research Center for Materials Nanoarchitectonics (MANA), National Institute for Materials Science (NIMS), Namiki 1-1, Tsukuba 305-0044, Ibaraki, Japan; liu.ran@sanken.osaka-u.ac.jp (R.L.); yamaura.kazunari@nims.go.jp (K.Y.); 2Graduate School of Chemical Sciences and Engineering, Hokkaido University, North 10 West 8, Kita-ku, Sapporo 060-0810, Hokkaido, Japan; 3Institute of Scientific and Industrial Research, Osaka University, Mihogaoka 8-1, Ibaraki 567-0047, Osaka, Japan; 4National Institute for Materials Science (NIMS), Sengen 1-2-1, Tsukuba 305-0047, Ibaraki, Japan; tanaka.masahiko@nims.go.jp

**Keywords:** A-site-ordered quadruple perovskites, B-site double ordering, crystal structures, structural disorder

## Abstract

ABO_3_ perovskite materials with small cations at the A site, especially with ordered cation arrangements, have attracted a lot of interest because they show unusual physical properties and deviations from general perovskite tendencies. In this work, A-site-ordered quadruple perovskites, RMn_3_Ni_2_Mn_2_O_12_ with R = Nd, Sm, Gd, and Dy, were synthesized by a high-pressure, high-temperature method at about 6 GPa. Annealing at about 1500 K produced samples with additional (partial) B-site ordering of Ni^2+^ and Mn^4+^ cations, crystallizing in space group *Pn*–3. Annealing at about 1700 K produced samples with disordering of Ni^2+^ and Mn^4+^ cations, crystallizing in space group *Im*–3. However, magnetic properties were nearly identical for the *Pn*–3 and *Im*–3 modifications in comparison with ferromagnetic double perovskites R_2_NiMnO_6_, where the degree of Ni^2+^ and Mn^4+^ ordering has significant effects on magnetic properties. In RMn_3_Ni_2_Mn_2_O_12_, one magnetic transition was found at 26 K (for R = Nd), 23 K (for R = Sm), and 22 K (for R = Gd), and two transitions were found at 10 K and 36 K for R = Dy. Curie–Weiss temperatures were close to zero in all compounds, suggesting that antiferromagnetic and ferromagnetic interactions are of the same magnitude.

## 1. Introduction

Perovskite-structure materials with a variety of cation orders attract a lot of attention, as properties can be tuned through different degrees of ordering. There are large subfamilies of perovskite-structure cation-ordered materials, for example, B-site double perovskites, A_2_B′B″O_6_ [1], and A-site-ordered quadruple perovskites, AA′_3_B_4_O_12_ [2,3,4,5,6,7]. There are more than one thousand examples of different A_2_B′B″O_6_ perovskites [1] and hundreds of AA′_3_B_4_O_12_ perovskites [2,3,4,5,6,7].

Among different possible combinations of B′ and B″ cations in A_2_B′B″O_6_, a combination of B′ = Ni^2+^ and B″ = Mn^4+^ attracts special attention as such a combination can produce ferromagnetic (FM) properties according to the Goodenough–Kanamori rules [8]. R_2_NiMnO_6_ perovskites, where R is a rare-earth element, were investigated a lot [9,10,11,12,13,14,15,16,17,18,19,20,21,22,23,24,25,26,27,28,29], and FM properties were indeed observed for all members from R = La (*T*_C_ ≈ 280 K [10,17,19]) to R = Lu (*T*_C_ ≈ 40 K [18,19,25,26,28]). The FM Curie temperature, *T*_C_, decreases in R_2_NiMnO_6_ as the deviation of the Ni–O–Mn bond angles increases from 180° [25,28]. In addition, the degree of Ni^2+^ and Mn^4+^ cation ordering in R_2_NiMnO_6_ has significant effects on the FM properties [9,18]. Near-room-temperature magnetocapacitance and magnetoresistance effects were observed in La_2_NiMnO_6_ [10,12,13].

The introduction of other cations [smaller than Lu^3+^ (*r*_XIII_(Lu^3+^) = 0.977 Å [30]), such as In^3+^ with *r*_XIII_(In^3+^) = 0.92 Å [30], Sc^3+^ with *r*_XIII_(Sc^3+^) = 0.870 Å [30], and Mn^2+^ with *r*_XIII_(Mn^2+^) = 0.96 Å [30])] into the A sites further reduces B–O–B′ bond angles and decreases the strength of direct B–B′ exchange interactions and can produce “exotic” properties [7,31]. For example, In_2_NiMnO_6_ already shows a complex incommensurate antiferromagnetic (AFM) ordering at *T*_N_ = 26 K and spin-induced ferroelectric polarization [21] in comparison with FM properties of R_2_NiMnO_6_. Sc_2_NiMnO_6_ demonstrates two AFM transitions and a complex magnetodielectric response [22]. Lu_2_NiMnO_6_ is located near a phase boundary, and external effects, such as moderate pressure, can induce a transition to an incommensurate AFM order from an FM order [25].

B-site double perovskites, A_2_B′B″O_6_, and A-site-ordered quadruple perovskites, AA′_3_B_4_O_12_, can be combined to simultaneously produce A-site- and B-site-ordered structures, AA′_3_B′_2_B″2O_12_ [32,33,34,35,36,37,38,39,40,41,42,43,44,45,46]. Depending on the combinations of B′ and B″ cations, such AA′_3_B’_2_B″2O_12_ perovskites can show large ferrimagnetic transitions above room temperature [35,37] and half-metallic properties [37,41]. In addition, such perovskites can show good catalytic properties [4], as they contain transition metals in different oxidation states and exotic magnetic ground states [28].

The R_2_NiMnO_6_ family of double perovskites was recently extended further through the synthesis of an AA′_3_B′_2_B″2O_12_-type perovskite, LaMn_3_Ni_2_Mn_2_O_12_ [28]. The average size of La^3+^ and 3Mn^3+^ cations is small; therefore, LaMn_3_Ni_2_Mn_2_O_12_ falls into the region with “exotic” properties, as Ni–O–Mn bond angles deviate significantly from 180° [28]. Two magnetic transitions were found in LaMn_3_Ni_2_Mn_2_O_12_ in comparison with other members of the R_2_NiMnO_6_ family (R = La–Lu), and complex magnetic structures were realized [28,29].

In this work, we prepared and investigated other members of RMn_3_Ni_2_Mn_2_O_12_ perovskites with R = Nd, Sm, Gd, and Dy, where the average size of R^3+^ and 3Mn^3+^ cations is further reduced (Table 1). In addition, we prepared two modifications of RMn_3_Ni_2_Mn_2_O_12_ perovskites with R = Nd and Sm, one with B-site ordering and the second without B-site ordering, and investigated the effects of B-site ordering on magnetic properties.

## 2. Results and Discussion

RMn_3_Ni_2_Mn_2_O_12_ samples with R = Nd, Sm, Gd, and Dy prepared at 1500 K crystallized in space group *Pn*–3 because of the observation of a (311) reflection on synchrotron XRPD data (Figure 1). The *Pn*–3 structure has two B sites and corresponds to an ordered arrangement of Ni^2+^ and Mn^4+^ cations (or at least partial ordering). The distribution of Ni^2+^ and Mn^4+^ cations between the two sites was refined with constraints on the full site occupation and the total chemical compositions. All the samples showed nearly the same refined occupation factors of 0.8Ni^2+^ + 0.2Mn^4+^ for the B site and 0.2Ni^2+^ + 0.8Mn^4+^ for the B′ site, suggesting a significant degree of Ni^2+^ and Mn^4+^ ordering. We also checked the occupation factors (*g*) of the R site and found that it was very close to 1 (*g*(Nd) = 1.0013(12), *g*(Sm) = 0.9994(11), *g*(Gd) = 1.0057(13), and *g*(Dy) = 1.0026(11)). Therefore, the occupation factor of the R site was fixed at 1 in the final models. All the samples contained a small amount of NiO impurity; the appearance of NiO impurity was also observed in LaMn_3_Ni_2_Mn_2_O_12_ [28]. The refined structural parameters and primary bond lengths for RMn_3_Ni_2_Mn_2_O_12_-*Pn*–3 are summarized in Table 2 and Table 3. Experimental, calculated, and difference synchrotron XRPD patterns are shown in Figure 1 for NdMn_3_Ni_2_Mn_2_O_12_-*Pn*–3 as an example.

The accuracy of determination of distributions of Ni^2+^ and Mn^4+^ cations with synchrotron XRPD is, of course, much lower than with neutron diffraction [28]. Nevertheless, with the obtained distributions of Ni^2+^ and Mn^4+^ cations, the refined isotropic atomic displacement parameters of the Ni_1_ and Mn_2_ sites were almost comparable to each other (about 0.4 Å^2^) for all compounds. On the other hand, the refined isotropic atomic displacement parameters were quite different for two extreme distributions of Ni^2+^ and Mn^4+^ cations: *B*(Ni_1_) = 0.78(3) Å^2^ and *B*(Mn_2_) = −0.05(2) Å^2^ for the full cation ordering and *B*(Ni_1_) = 0.11(2) Å^2^ and *B*(Mn_2_) = 0.65(3) Å^2^ for the full cation disordering (for the R = Nd sample as an example).

RMn_3_Ni_2_Mn_2_O_12_ samples with R = Nd and Sm prepared at 1700 K crystallized in space group *Im*–3 because of the absence of a (311) reflection on synchrotron XRPD data (Figure 2). The *Im*–3 structure has one B site and, therefore, corresponds to a disordered arrangement of Ni^2+^ and Mn^4+^ cations. The refined structural parameters and primary bond lengths for RMn_3_Ni_2_Mn_2_O_12_-*Im*–3 are summarized in Table 4. Experimental, calculated, and difference synchrotron XRPD patterns are shown in Figure 2 for NdMn_3_Ni_2_Mn_2_O_12_-*Im*–3 as an example. RMn_3_Ni_2_Mn_2_O_12_ samples with R = Nd and Sm prepared at 1700 K crystallized in space group *Im*–3 because of the absence of a (311) reflection on synchrotron XRPD data (Figure 2). The *Im*–3 structure has one B site and, therefore, corresponds to the disordered arrangement of Ni^2+^ and Mn^4+^ cations. Experimental, calculated, and difference synchrotron XRPD patterns are shown in Figure 2 for NdMn_3_Ni_2_Mn_2_O_12_-*Im*–3 as an example.

Figure 3 shows lattice parameters as a function of the ionic radius [30] for RMn_3_Ni_2_Mn_2_O_12_ samples with R = La [28], Nd, Sm, Gd, and Dy. Nearly linear behavior of the cubic lattice parameter was observed. Figure 3 also shows R–O bond lengths and bond valence sum (BVS) values [47] as a function of the ionic radius. The BVS values of Mn^3+^, Mn^4+^, and Ni^2+^ sites/cations were almost constant in all RMn_3_Ni_2_Mn_2_O_12_-*Pn*–3 compounds independent of the R^3+^ cations and agreed well with the expected formal oxidation states. On the other hand, the R–O bond length and the BVS values of the R sites changed monotonically and noticeably, and they were dependent on the R^3+^ cations. In LaMn_3_Ni_2_Mn_2_O_12_ [28], the BVS value of the La site was +3.40, indicating that La^3+^ is highly overbonded (probably for this reason, the BVS value of the La site was not mentioned and discussed in [28]). In DyMn_3_Ni_2_Mn_2_O_12_, the BVS value of the Dy site was +2.72, indicating that Dy^3+^ is noticeably underbonded. The optimal BVS value of +3.0 is realized in SmMn_3_Ni_2_Mn_2_O_12_. Severe underbonding of R^3+^ could be a reason why RMn_3_Ni_2_Mn_2_O_12_ with smaller R^3+^ cations (R = Er and Tm) could not be prepared.

Magnetic properties of NdMn_3_Ni_2_Mn_2_O_12_ and SmMn_3_Ni_2_Mn_2_O_12_ compounds in two modifications (*Pn*–3 and *Im*–3) are shown in Figure 4 and Figure 5 (also marked by the synthesis temperatures between 1500 K and 1700 K). Temperature-dependent magnetic properties were almost identical for the two modifications. Magnetic properties of GdMn_3_Ni_2_Mn_2_O_12_ and DyMn_3_Ni_2_Mn_2_O_12_ compounds (the *Pn*–3 modification) are shown in Figure 6 and Figure 7; they were dominated by large moments of Gd^3+^ and Dy^3+^ cations. Nevertheless, differential *dχ*/*dT* versus *T* curves allowed the detection of one magnetic anomaly in GdMn_3_Ni_2_Mn_2_O_12_ and two magnetic anomalies in DyMn_3_Ni_2_Mn_2_O_12_. At high temperatures, inverse magnetic susceptibilities followed the Curie–Weiss law (see Figure 7 as an example), and we obtained Curie–Weiss fitting parameters using the 10 kOe FCC curves in a temperature range of 200–350 K. The Curie–Weiss fitting parameters are summarized in Table 5. The experimental effective magnetic moments (*μ*_eff_) were close to the expected calculated values (*μ*_calc_). It is interesting that the Curie–Weiss temperatures (*θ*) were very small, suggesting that antiferromagnetic and ferromagnetic interactions are of the same magnitude and nearly cancel each other. In GdMn_3_Ni_2_Mn_2_O_12_ and DyMn_3_Ni_2_Mn_2_O_12_ compounds, the Curie–Weiss temperatures were slightly positive, suggesting that ferromagnetic interactions are slightly stronger. Unfortunately, no Curie–Weiss fits were reported for LaMn_3_Ni_2_Mn_2_O_12_ [28] to compare with our data. It is possible that a similar “strange” Curie–Weiss temperature was obtained; therefore, such data were not mentioned and discussed. In all R_2_NiMnO_6_ (R = La–Lu [10,16,17,18,19]), Tl_2_NiMnO_6_ [23], and In_2_NiMnO_6_ [20], positive and large Curie–Weiss temperatures were found indicating that ferromagnetic interactions are dominant. On the other hand, in Sc_2_NiMnO_6_ [22], a negative Curie–Weiss temperature of about −60 K was observed, indicating that antiferromagnetic interactions are dominant. Therefore, there could be strong competition of different magnetic interactions in RMn_3_Ni_2_Mn_2_O_12,_ resulting in nearly zero Curie–Weiss temperatures.

Isothermal magnetization curves (M versus H) of NdMn_3_Ni_2_Mn_2_O_12_ and SmMn_3_Ni_2_Mn_2_O_12_ compounds in two modifications (*Pn*–3 and *Im*–3) are shown in Figure 8. Again, M versus H curves were almost identical for the two modifications. Small hysteresis near the origin was found, but no saturation behavior was observed. M versus H curves of GdMn_3_Ni_2_Mn_2_O_12_ and DyMn_3_Ni_2_Mn_2_O_12_ compounds (*Pn*–3 modifications) are shown in Figure 9; small hysteresis near the origin was observed in GdMn_3_Ni_2_Mn_2_O_12_, while no hysteresis was detected in DyMn_3_Ni_2_Mn_2_O_12_. The M versus H curves of DyMn_3_Ni_2_Mn_2_O_12_ were mainly determined by properties of Dy^3+^ cations and suggest the absence of any ferromagnetic-like contributions. Except for a very weak, extended hysteresis in SmMn_3_Ni_2_Mn_2_O_12_, its M versus H curve also suggests the absence of any significant ferromagnetic-like contributions. On the other hand, M versus H curves of NdMn_3_Ni_2_Mn_2_O_12_ and GdMn_3_Ni_2_Mn_2_O_12_ were similar to those of LaMn_3_Ni_2_Mn_2_O_12_ [28] and suggest the presence of ferromagnetic-like contributions.

Specific heat data for two modifications of SmMn_3_Ni_2_Mn_2_O_12_ were measured (Figure 10b), and almost no difference was observed. Specific heat data for the *Pn*–3 modification of other compounds (R = Nd, Gd, and Dy) are shown in Figure 10. Specific heat measurements confirmed one clear magnetic transition in the samples with R = Nd, Sm, and Gd and two clear magnetic transitions in the sample with R = Dy in agreement with the *χ* versus *T* measurements. In the case of LaMn_3_Ni_2_Mn_2_O_12_ [28], specific heat measurements could detect two magnetic transitions at 34 K and 46 K, where the transition at 46 K was assigned to a long-range ordering of Mn^3+^ cations at the square-planar A′ sites, and the transition at 34 K was assigned to a long-range ordering of Ni^2+^ and Mn^4+^. Transition temperatures found in R = Nd, Sm, and Gd were noticeably smaller than those of R = La. Therefore, it is possible that the size of R^3+^ cations plays an important role and can move the systems into different ground states. Another possibility is that the degree of Ni^2+^ and Mn^4+^ cation ordering plays an important role and determines magnetic transition temperatures.

To get a deeper understanding of magnetic behavior, we measured ac magnetic susceptibility curves of the *Pn*–3 modification of NdMn_3_Ni_2_Mn_2_O_12_ as an example (Figure 11). Small frequency dependence in the χ″ peak intensities was observed; however, peak positions in temperature were almost independent of frequency, suggesting that there are no spin-glass-like contributions. No dependence on the applied *H*_ac_ field of 0.05, 0.5, and 5 Oe (inset of Figure 11) was also observed on both the χ′ versus *T* and the χ″ versus *T* curves. Peaks on the χ′ versus *T* curves were observed near 14 K, while peaks on the χ″ versus *T* curves were observed near 9 K. Both temperatures were different from peak positions on the dc d*χT*/d*T* versus *T* curves, but peaks on the χ′ versus *T* curves basically matched with the peak on the ZFC dc *χ* versus *T* curve measured at a small magnetic field of *H* = 100 Oe (Figure 4b).

Considering the presence of spin-induced ferroelectric properties in In_2_NiMnO_6_ [21] and complex magnetodielectric effects in Sc_2_NiMnO_6_ [22], we checked the presence or absence of magnetodielectric effects in such RMn_3_Ni_2_Mn_2_O_12_ perovskites selecting the *Pn*–3 modification of NdMn_3_Ni_2_Mn_2_O_12_ as an example (Figure 12). However, no dielectric anomalies were observed near the magnetic transition temperature. The dielectric constant showed a sharp increase above about 50 K (at 100 Hz) with characteristic frequency-dependent peaks on the loss tangent. These features correspond to increased conductivity and Maxwell–Wagner contributions. Therefore, NdMn_3_Ni_2_Mn_2_O_12_ perovskite does not show magnetodielectric effects and magnetic-transition-induced ferroelectric polarization.

In the case of double perovskites R_2_NiMnO_6_ with ferromagnetic ground states, the degree of ordering of Ni^2+^ and Mn^4+^ cations has significant effects on magnetic properties, especially on the saturation magnetization on M versus H curves, where a full ordering of Ni^2+^ and Mn^4+^ cations should give the magnetization of about 5*μ*_B_/f.u. [10]. Deviations of the saturation magnetization from 5*μ*_B_/f.u. can give information about the degree of Ni^2+^ and Mn^4+^ disordering. On the other hand, we found that the saturation magnetization on the M versus H curves of RMn_3_Ni_2_Mn_2_O_12_ (and other magnetic properties) was almost independent of the degree of Ni^2+^ and Mn^4+^ disordering. This fact suggests that different magnetic ground states could be realized in all RMn_3_Ni_2_Mn_2_O_12_ in comparison with R_2_NiMnO_6_, and magnetic structures of RMn_3_Ni_2_Mn_2_O_12_ should be studied with neutron diffraction in future works. Magnetic transition temperatures and properties did not show any clear systematic trends as a function of the ionic radius of R^3+^ cations in RMn_3_Ni_2_Mn_2_O_12_. On the other hand, in R_2_NiMnO_6_ perovskites, magnetic transition temperatures show a clear and sharp decrease with decreasing the ionic radius of R^3+^ cations [18,19,25,28].

Effects of the synthesis conditions on the degree of B-site cation ordering were observed before, for example, in La_2_NiMnO_6_ [9], Tl_2_NiMnO_6_ [23], and CaCu_3_Fe_2_Os_2_O_12_ [36], where higher-temperature annealing (at high pressures) usually results in B-site cation disordering. Our results on RMn_3_Ni_2_Mn_2_O_12_ are consistent with the tendencies observed in the literature.

## 3. Materials and Methods

RMn_3_Ni_2_Mn_2_O_12_ samples with R = Nd, Sm, Gd, and Dy were prepared from stoichiometric mixtures of R_2_O_3_ (Rare Metallic Co., Tokyo, Japan, 99.9%), Mn_2_O_3_, single-phase and stoichiometric MnO_2_ (Alfa Aesar, Waltham, MA, USA, 99.9%), and NiO (Rare Metallic Co., Tokyo, Japan, 99.9%). Single-phase Mn_2_O_3_ was prepared from a commercial MnO_2_ chemical (Rare Metallic Co., Tokyo, Japan, 99.99%) by annealing in air at 923 K for 24 h. The synthesis was performed at about 6 GPa and at about 1500 K for 2 h in sealed Au capsules and at about 1700 K for 2 h in sealed Pt capsules using a belt-type HP instrument. After annealing at high temperatures, the samples were cooled down to room temperature by turning off the heating current, and the pressure was slowly released. We note that we also tried to prepare RMn_3_Ni_2_Mn_2_O_12_ samples with smaller R^3+^ cations, such as R = Er and Tm (at 6 GPa and 1500 K). However, the samples contained a lot of impurities. The R = Er sample (with *a* = 7.3189 Å) had ErMn_2_O_5_ (9.5 wt. %), NiO (9.2 wt. %), and a corundum-structure impurity (2.4 wt. %). The R = Tm sample (with *a* = 7.3190 Å) had TmMn_2_O_5_ (15.7 wt. %), NiO (8.2 wt. %), and a corundum-structure impurity (6.1 wt. %). The presence of large amounts of impurities suggests that the chemical compositions of the main phases significantly shifted from the target composition.

X-ray powder diffraction (XRPD) data were collected at room temperature on a MiniFlex600 diffractometer (Rigaku, Tokyo, Japan) using CuKα radiation (2*θ* range of 8–100°, a step width of 0.02°, and scan speed of 2°/min). Room-temperature synchrotron XRPD data were measured on the BL15XU beamline (the former NIMS beamline) of SPring-8 (Hyogo, Japan) [48] between 2.04° and 60.23° at 0.003° intervals in 2*θ* with the wavelength of *λ* = 0.65298 Å. The samples were placed into open Lindemann glass capillary tubes (inner diameter: 0.1 mm), which were rotated during measurements. The Rietveld analysis of all XRPD data was performed using the RIETAN-2000 program [49].

Magnetic measurements were performed on SQUID magnetometers (Quantum Design MPMS-7T and MPMS3, San Diego, CA, USA) between 2 and 300 K (or 400 K) in applied fields of 100 Oe and 10 kOe under both zero-field-cooled (ZFC) and field-cooled on cooling (FCC) conditions. Magnetic-field dependence was measured at different temperatures between −70 and 70 kOe. Frequency-dependent alternating current (ac) susceptibility measurements were performed on cooling with an MPMS-1T instrument Quantum Design, San Diego, CA, USA) at different frequencies (*f*), different applied oscillating magnetic fields (*H*_ac_), and zero static dc field (*H*_dc_ = 0 Oe).

Specific heat, *C*_p_, was measured on cooling from 270 K to 2 K at zero magnetic field and from 100 K to 2 K at a magnetic field of 90 kOe by a pulse relaxation method using a commercial calorimeter (Quantum Design PPMS, San Diego, CA, USA). All magnetic and specific heat measurements were performed using pieces of pellets.

Dielectric properties were measured using an Alpha-A High-Performance Frequency Analyzer (NOVOCONTROL Technologies, Montabaur, Germany) on cooling and heating in a temperature range between 3 K and 300 K and a frequency range from 100 Hz to 665 kHz at a zero magnetic field. The cooling–heating rate was 2 K/min between 70 K and 300 K and 0.5 K/min between 3 K and 70 K.

Scanning electron microscopy (SEM) images were obtained on a Miniscope TM3000 operating at 15 kV (Hitachi, Tokyo, Japan).

## 4. Conclusions

A-site-ordered quadruple perovskites, RMn_3_Ni_2_Mn_2_O_12_ with R = Nd, Sm, Gd, and Dy, were synthesized by a high-pressure, high-temperature method at about 6 GPa in two modifications. Annealing at a lower temperature of about 1500 K favors a (partial) B-site ordering, while annealing at a higher temperature of about 1700 K gives a disordered arrangement of Ni^2+^ and Mn^4+^ cations. The B-site-ordered structure has space group *Pn*–3, while the B-site-disordered structure has space group *Im*–3. However, magnetic properties were nearly identical for the *Pn*–3 and *Im*–3 modifications in comparison with ferromagnetic double perovskites R_2_NiMnO_6_. RMn_3_Ni_2_Mn_2_O_12_ samples show one magnetic transition at 26 K for R = Nd, 23 K for R = Sm, and 22 K for R = Gd, as well as two magnetic transitions at 10 K and 36 K for R = Dy. Curie–Weiss temperatures were close to zero in all compounds, suggesting that antiferromagnetic and ferromagnetic interactions are of the same magnitude.

## Figures and Tables

**Figure 1 molecules-29-05488-f001:**
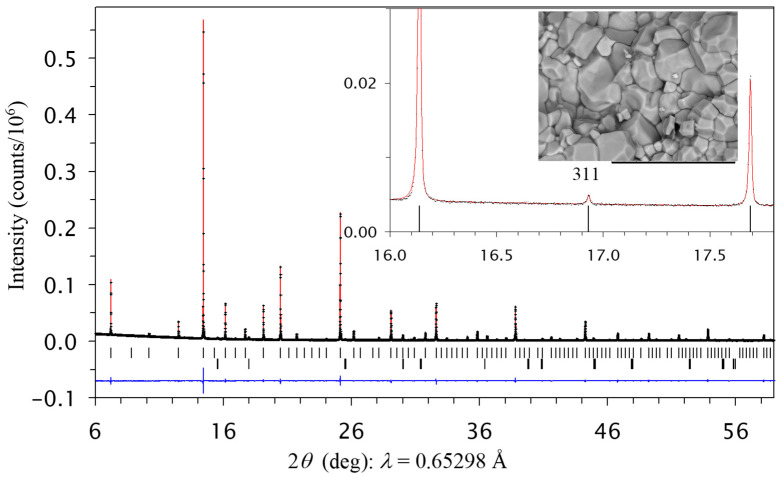
Experimental (black crosses), calculated (red line), and difference (blue line at the bottom) room-temperature synchrotron X-ray powder diffraction patterns of NdMn_3_Ni_2_Mn_2_O_12_ (in the *Pn*–3 modification, prepared at 1500 K) in a 2*θ* range of 6° and 59°. The tick marks show possible Bragg reflection positions for the main phase and NiO impurity (from top to bottom). Inset shows a zoomed part in a 2*θ* range of 16° and 17.8° and emphasizes the presence of the (311) reflection from the B-site ordering. Inset shows a scanning electron microscopy (SEM) image, where the scale bar is 20 µm.

**Figure 2 molecules-29-05488-f002:**
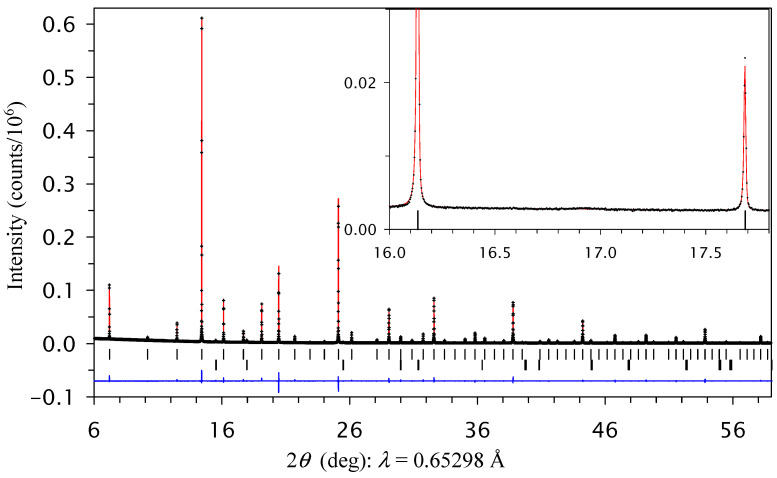
Experimental (black crosses), calculated (red line), and difference (blue line at the bottom) room-temperature synchrotron X-ray powder diffraction patterns of NdMn_3_Ni_2_Mn_2_O_12_ (in the *Im*–3 modification, prepared at 1700 K) in a 2*θ* range of 6° and 59°. The tick marks show possible Bragg reflection positions for the main phase and NiO impurity. Inset shows a zoomed part in a 2*θ* range of 16° and 17.9° and emphasizes the absence of the (311) reflection and the absence of B-site ordering.

**Figure 3 molecules-29-05488-f003:**
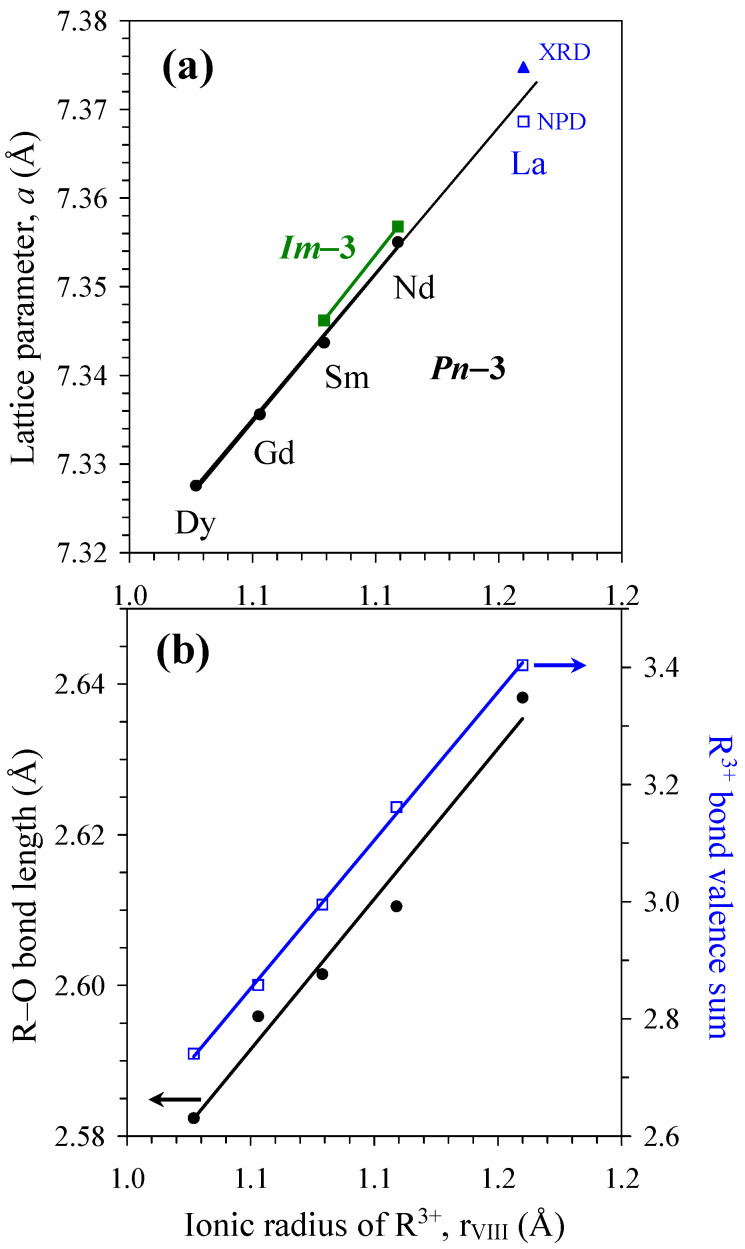
(**a**) The room-temperature cubic lattice parameter in RMn_3_Ni_2_Mn_2_O_12_ (R = La [28], Nd, Sm, Gd, and Dy) as a function of the ionic radius R^3+^ (for the coordination number 8 as ionic radii for the coordination number XII are not available for small R^3+^ cations (R = Gd and Dy) [30]). NPD: from neutron powder diffraction. XRD: from X-ray powder diffraction. (**b**) R–O bond length (the left-hand axis) and bond-valence sum for R^3+^ (the right-hand axis) in RMn_3_Ni_2_Mn_2_O_12_ (R = La [28], Nd, Sm, Gd, and Dy) as a function of the ionic radius R^3+^.

**Figure 4 molecules-29-05488-f004:**
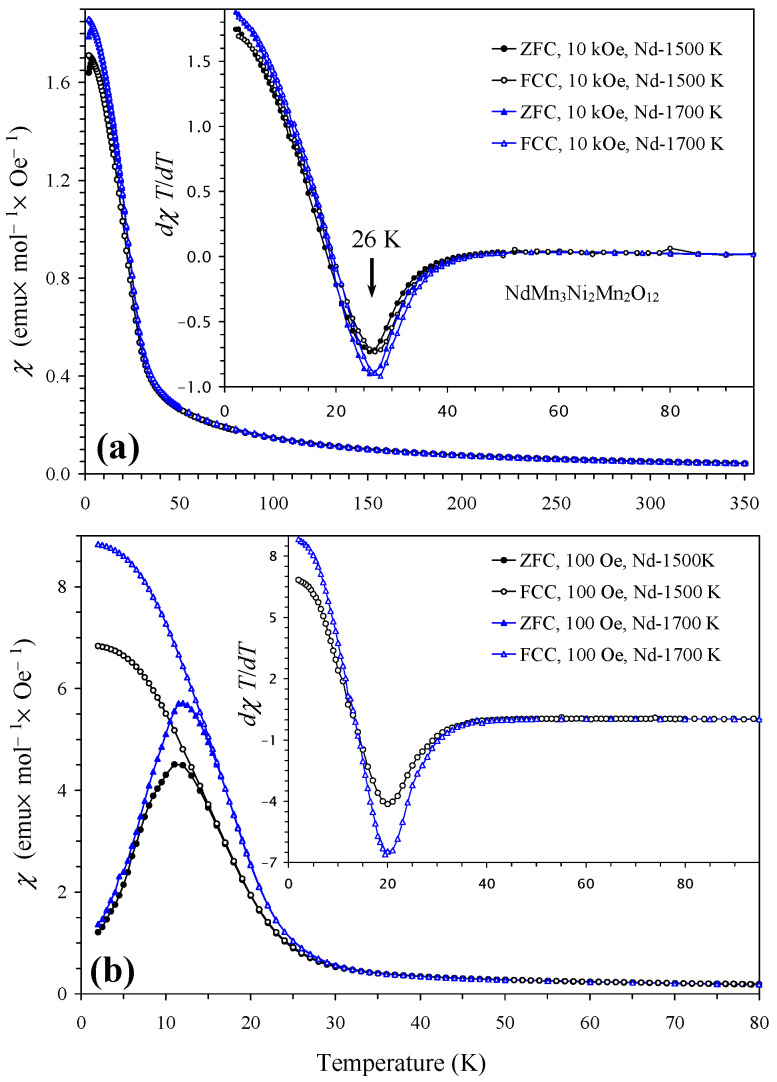
(**a**) ZFC (filled symbols) and FCC (empty symbols) dc magnetic susceptibility curves (*χ* = *M*/*H*) of two modifications of NdMn_3_Ni_2_Mn_2_O_12_ (the *Pn*–3 modification, prepared at 1500 K, and the *Im*–3 modification, prepared at 1700 K) measured at *H* = 10 kOe. The inset shows the d*χT*/d*T* versus *T* curves (all). (**b**) ZFC and FCC curves of two modifications of NdMn_3_Ni_2_Mn_2_O_12_ measured at *H* = 100 Oe. The inset shows the FCC d*χT*/d*T* versus *T* curves.

**Figure 5 molecules-29-05488-f005:**
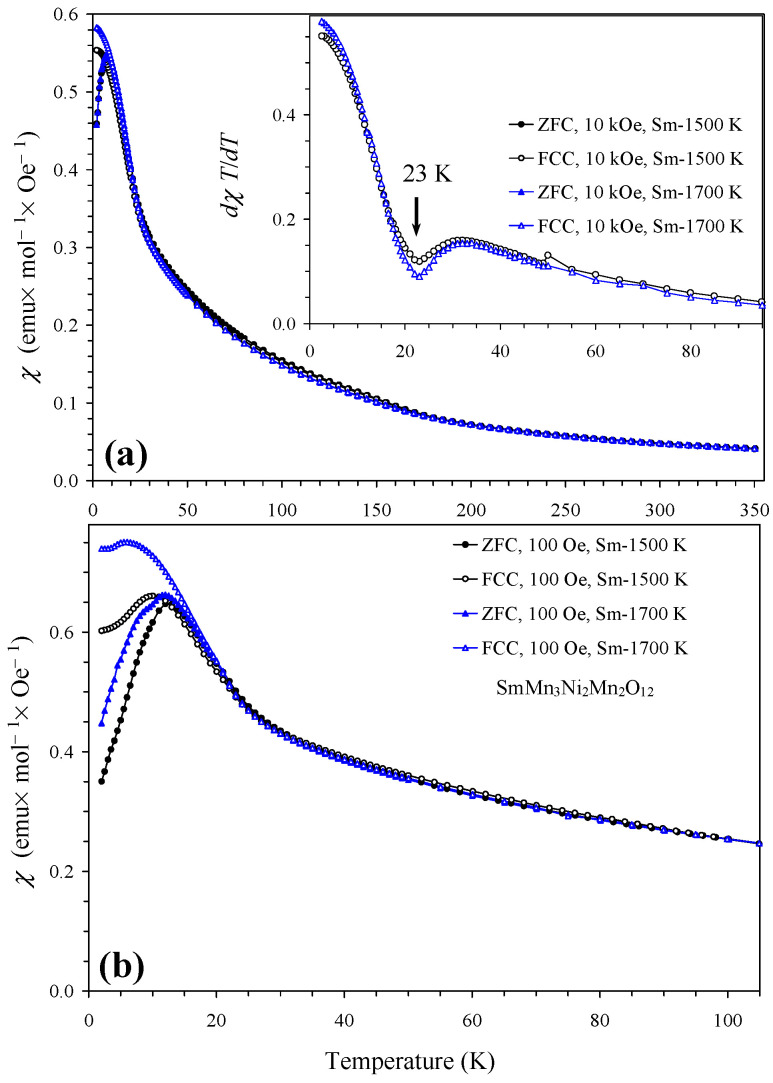
(**a**) ZFC (filled symbols) and FCC (empty symbols) dc magnetic susceptibility curves (*χ* = *M*/*H*) of two modifications of SmMn_3_Ni_2_Mn_2_O_12_ (the *Pn*–3 modification, prepared at 1500 K, and the *Im*–3 modification, prepared at 1700 K) measured at *H* = 10 kOe. The inset shows FCC d*χT*/d*T* versus *T* curves. (**b**) ZFC and FCC curves of two modifications of SmMn_3_Ni_2_Mn_2_O_12_ measured at *H* = 100 Oe.

**Figure 6 molecules-29-05488-f006:**
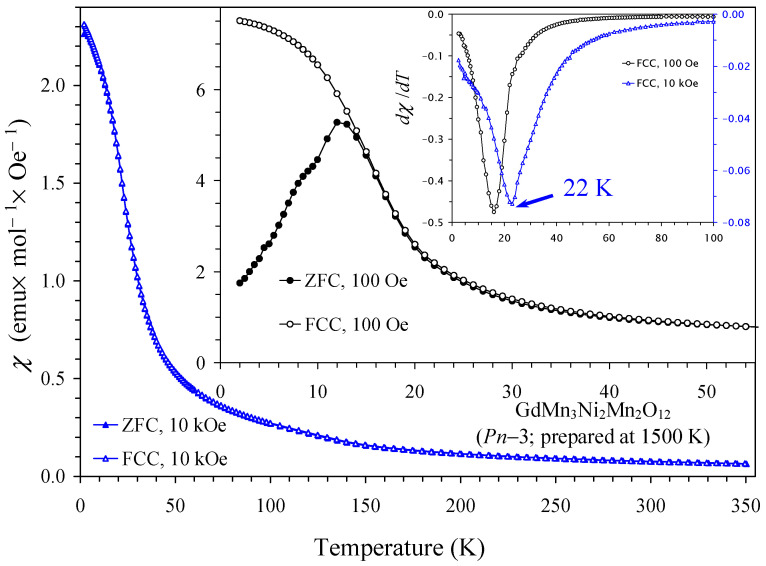
ZFC (filled symbols) and FCC (empty symbols) dc magnetic susceptibility curves (*χ* = *M*/*H*) of GdMn_3_Ni_2_Mn_2_O_12_ (the *Pn*–3 modification, prepared at 1500 K) measured at *H* = 10 kOe. The first inset shows ZFC and FCC curves of GdMn_3_Ni_2_Mn_2_O_12_ measured at *H* = 100 Oe. The second inset shows the FCC d*χ*/d*T* versus *T* curves.

**Figure 7 molecules-29-05488-f007:**
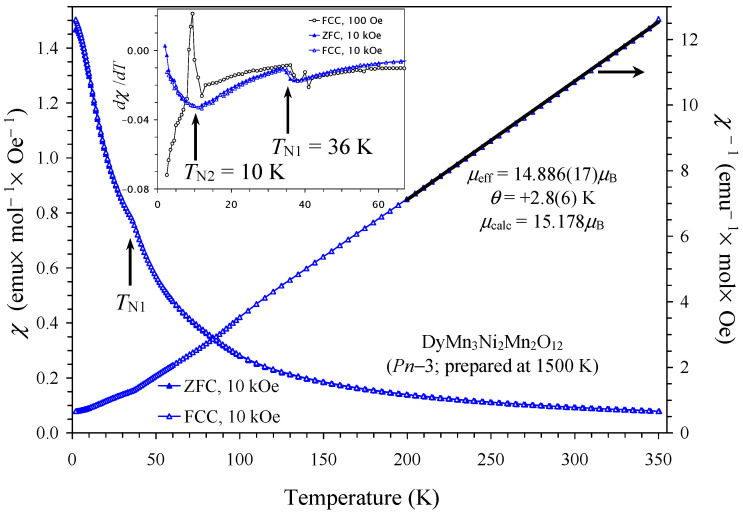
ZFC (filled symbols) and FCC (empty symbols) dc magnetic susceptibility curves (*χ* = *M*/*H*) of DyMn_3_Ni_2_Mn_2_O_12_ (the *Pn*–3 modification, prepared at 1500 K) measured at *H* = 10 kOe (the left-hand axis). The right-hand axis shows the FCC *χ*^−1^ versus *T* curve with the Curie–Weiss fit (black line). The fitting parameters are given in the figure. The inset shows d*χ*/d*T* versus *T* curves.

**Figure 8 molecules-29-05488-f008:**
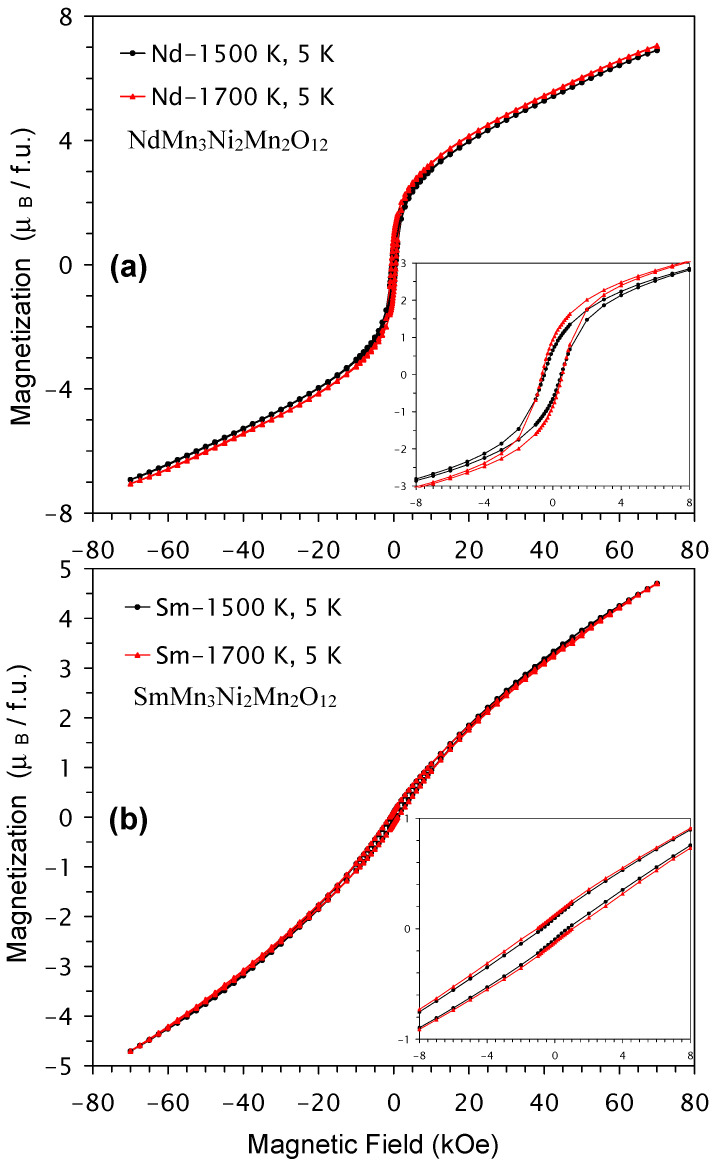
M versus H curves of two modifications of RMn_3_Ni_2_Mn_2_O_12_ (the *Pn*–3 modification, prepared at 1500 K, and the *Im*–3 modification, prepared at 1700 K) measured at *T* = 5 K with (**a**) R = Nd and (**b**) R = Sm. The insets show zoomed parts near the origin.

**Figure 9 molecules-29-05488-f009:**
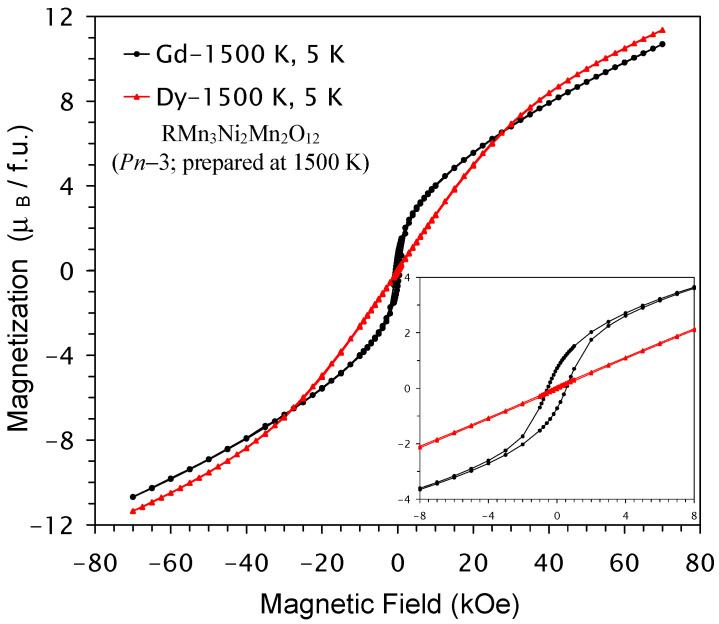
M versus H curves of GdMn_3_Ni_2_Mn_2_O_12_ and DyMn_3_Ni_2_Mn_2_O_12_ (the *Pn*–3 modification, prepared at 1500 K) measured at *T* = 5 K. The inset shows zoomed parts near the origin.

**Figure 10 molecules-29-05488-f010:**
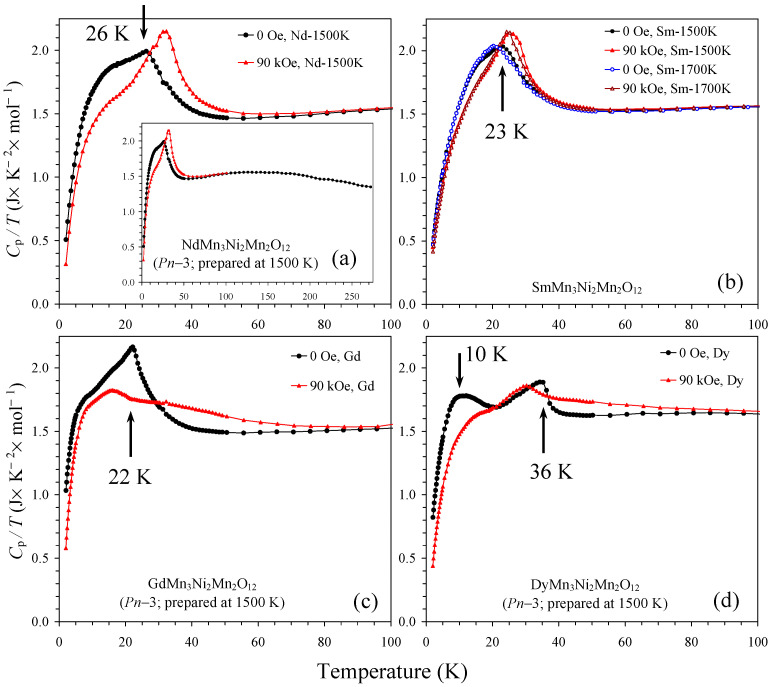
*C*_p_/*T* versus *T* curves of RMn_3_Ni_2_Mn_2_O_12_ measured at *H* = 0 (black curves) and 90 kOe (red curves) for (**a**) R = Nd (the *Pn*–3 modification), (**b**) R = Sm (the *Pn*–3 modification and the *Im*–3 modification (blue and brown curves)), (**c**) R = Gd (the *Pn*–3 modification), and (**d**) R = Dy (the *Pn*–3 modification). Arrows show magnetic transition temperatures. Data below 100 K are shown; inset on panel (**a**) shows full data up to 270 K (at *H* = 0 Oe).

**Figure 11 molecules-29-05488-f011:**
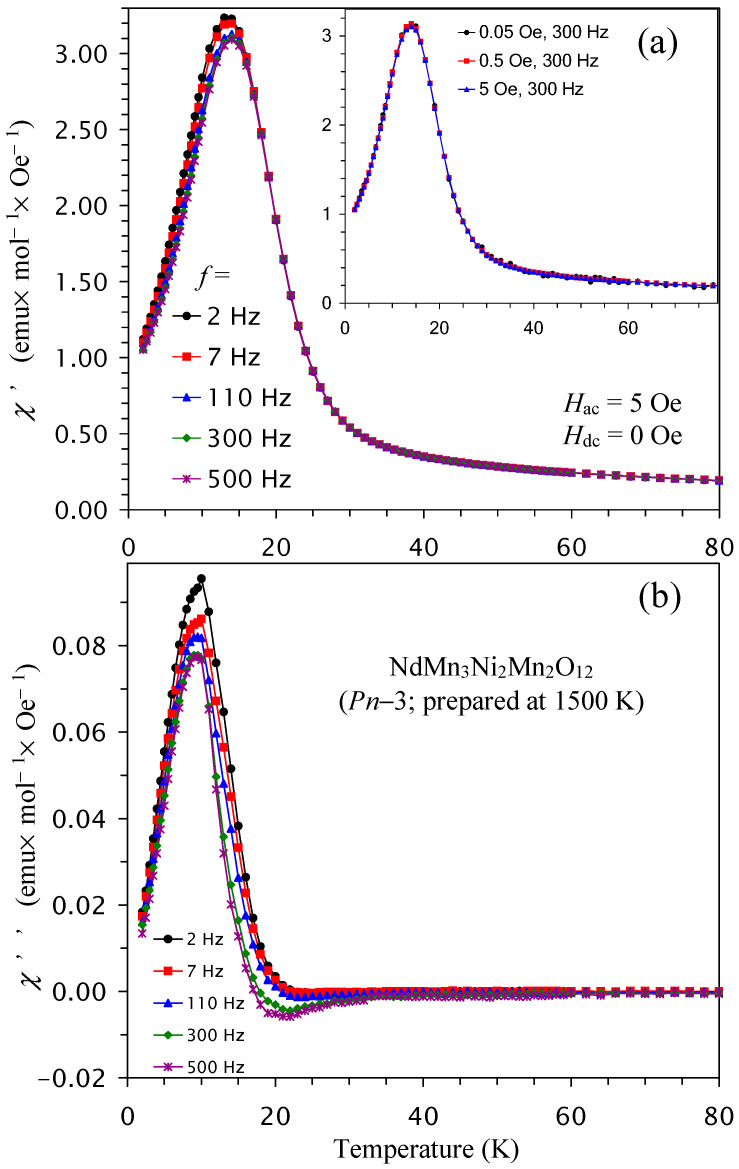
(**a**) Real χ′ versus *T* and (**b**) imaginary χ″ versus *T* curves of NdMn_3_Ni_2_Mn_2_O_12_ (the *Pn*–3 modification) at different frequencies (*f*). Inset shows the χ′ versus *T* curves at different *H*_ac_ (*H*_ac_ = 0.05, 0.5, and 5 Oe) and one frequency (*f* = 300 Hz).

**Figure 12 molecules-29-05488-f012:**
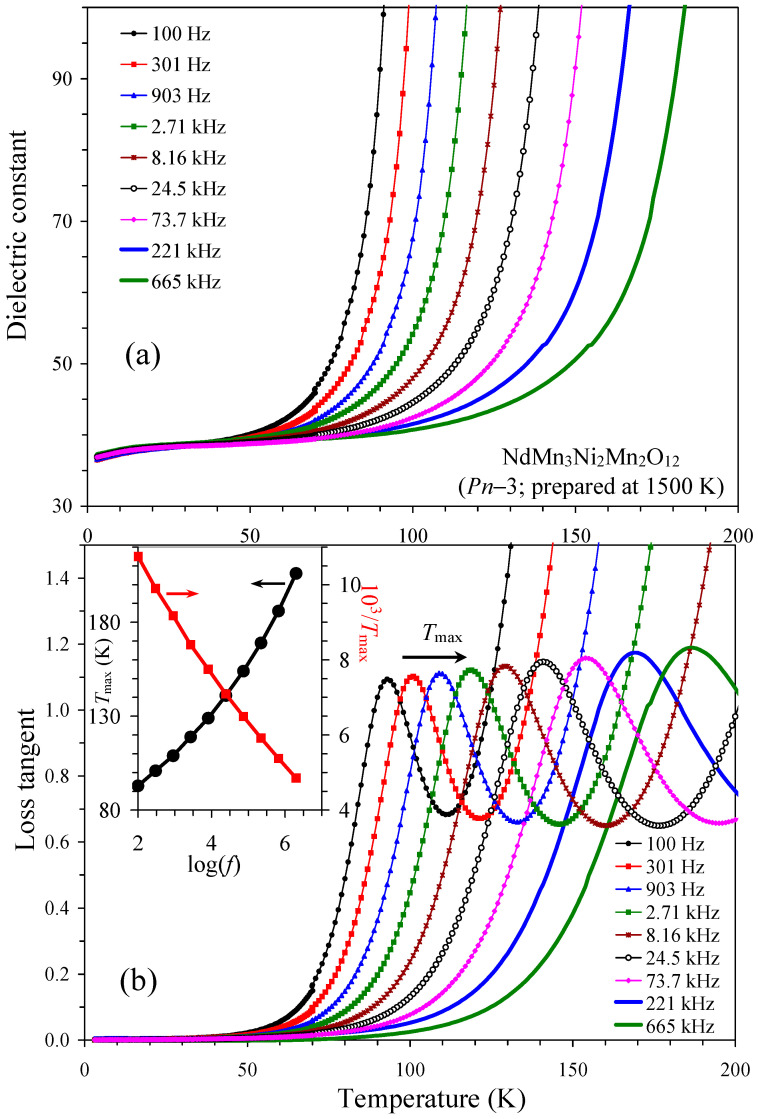
Temperature dependence of (**a**) dielectric constant and (**b**) loss tangent at different frequencies (*f*: indicated on the figure) in NdMn_3_Ni_2_Mn_2_O_12_ (the *Pn*–3 modification) at *H* = 0 Oe. Inset shows frequency dependence of peak positions on loss tangent as *T*_max_ versus log(*f*) (black circles with line) and 1000/*T*_max_ versus log(*f*) (red squares with line).

**Table 1 molecules-29-05488-t001:** A list of RMn_3_Ni_2_Mn_2_O_12_ samples prepared (at a high pressure of 6 GPa) and investigated in this work.

R	Synthesis Temperature	Symmetry
Nd	1500 K	*Pn*–3
Nd	1700 K	*Im*–3
Sm	1500 K	*Pn*–3
Sm	1700 K	*Im*–3
Gd	1500 K	*Pn*–3
Dy	1500 K	*Pn*–3

**Table 2 molecules-29-05488-t002:** Structure parameters of RMn_3_Ni_2_Mn_2_O_12_ (*Pn*–3; prepared at 1500 K) at room temperature from synchrotron powder X-ray diffraction data.

R	Nd	Sm	Gd	Dy
*a* (Å)	7.35504(1)	7.34371(1)	7.33561(1)	7.32757(1)
*V* (Å^3^)	397.8824(2)	396.0468(5)	394.7375(5)	393.4419(4)
*B*_iso_(R) (Å^2^)	0.436(5)	0.532(5)	0.551(6)	0.661(5)
*B*_iso_(Mn_SQ_) (Å^2^)	0.541(9)	0.578(8)	0.560(9)	0.550(7)
*g*(Ni_1_/Mn_1_)	0.82(2)Ni+ 0.18Mn	0.834(17)Ni+ 0.166Mn	0.82(2)Ni+ 0.18Mn	0.831(16)Ni+ 0.169Mn
*B*_iso_(Ni_1_/Mn_1_) (Å^2^)	0.36(4)	0.42(4)	0.34(4)	0.38(3)
*g*(Mn_2_/Ni_2_)	0.82Mn+ 0.18Ni	0.834Mn+ 0.166Ni	0.82Mn+ 0.18Ni	0.831Mn+ 0.169Ni
*B*_iso_(Mn_2_/Ni_2_) (Å^2^)	0.38(4)	0.36(4)	0.45(4)	0.34(4)
*x*(O)	0.2576(5)	0.2576(5)	0.2576(6)	0.2574(5)
*y*(O)	0.42527(16)	0.42542(16)	0.42462(19)	0.42412(16)
*z*(O)	0.55854(15)	0.55767(14)	0.55771(17)	0.55632(14)
*B*_iso_(O) (Å^2^)	0.49(3)	0.62(3)	0.54(3)	0.60(3)
*R*_wp_ (%)	3.66	4.17	4.66	4.85
*R*_p_ (%)	2.40	2.78	2.87	3.32
*R*_p_ (%)	1.89	2.44	1.94	2.26
*R*_F_ (%)	2.42	3.02	2.55	3.13
Impurities:				
NiO (*R*–3*m*)	3.0 wt. %	2.8 wt. %	1.5 wt. %	1.8 wt. %
GdFeO_3_-related	–	0.5 wt. %	2.0 wt. %	1.7 wt. %

Space group *Pn*–3 (No. 201, setting 2); *Z* = 2. Fractional coordinates: R: 2*a* (0.25, 0.25, 0.25), Mn_SQ_: 6*d* (0.25, 0.75, 0.75), Ni_1_/Mn_1_: 4*b* (0, 0, 0), Mn_2_/Ni_2_: 4*c* (0.5, 0.5, 0.5), and O: 24*h* (*x*, *y*, *z*). Occupation factors, *g*, of the R, Mn_SQ_, O sites are 1. Constraints on occupation factors: *g*(Mn_1_) = *g*(Ni_2_) = 1 − *g*(Ni_1_) and *g*(Mn_2_) = *g*(Ni_1_). GdFeO_3_-related impurities: for R = Sm, space group *Pnma*, *a* = 5.5165 Å, *b* = 7.6078 Å, *c* = 5.3470 Å; for R = Gd, space group *P*2_1_/*n*, *a* = 5.2908 Å, *b* = 5.5452 Å, *c* = 7.5560 Å, *β* = 90.1356°; for R = Dy, space group *P*2_1_/*n*, *a* = 5.2452 Å, *b* = 5.5423 Å, *c* = 7.4960 Å, *β* = 90.2082°.

**Table 3 molecules-29-05488-t003:** Bond lengths (in Å), bond valence sums (BVS), and Ni–O–Mn bond angles (in deg) in RMn_3_Ni_2_Mn_2_O_12_ (*Pn*–3; prepared at 1500 K) at room temperature from synchrotron powder X-ray diffraction data.

R	Nd	Sm	Gd	Dy
R–O × 12 (Å)	2.6105(11)	2.6015(11)	2.5959(13)	2.5824(11)
BVS(R^3+^)	3.16	3.00	2.86	2.74
Mn_SQ_–O × 4 (Å)	1.9099(11)	1.9125(11)	1.9062(14)	1.9091(11)
Mn_SQ_–O × 4 (Å)	2.7732(12)	2.7712(11)	2.7731(14)	2.7784(11)
BVS(Mn^3+^)	2.93	2.91	2.95	2.93
Ni_1_/Mn_1_–O × 6 (Å)	2.019(4)	2.014(3)	2.014(4)	2.009(3)
BVS(Ni^2+^)	2.24	2.27	2.27	2.30
Mn_2_/Ni_2_–O × 6 (Å)	1.915(4)	1.910(3)	1.910(4)	1.908(3)
BVS(Mn^4+^)	3.87	3.92	3.93	3.95
Ni_1_–O–Mn_2_	138.38(6)	138.65(6)	138.38(8)	138.56(6)

**Table 4 molecules-29-05488-t004:** Structure parameters, bond lengths, and bond valence sums (BVSs) of RMn_3_Ni_2_Mn_2_O_12_ (*Im*–3; prepared at 1700 K) at room temperature from synchrotron powder X-ray diffraction data.

R	Nd	Sm
*a* (Å)	7.35677(1)	7.34621(1)
*V* (Å^3^)	398.1629(5)	396.4512(5)
*B*_iso_(R) (Å^2^)	0.382(5)	0.285(5)
*B*_iso_(Mn_SQ_) (Å^2^)	0.558(9)	0.413(8)
*B*_iso_(Ni/Mn) (Å^2^)	0.297(7)	0.228(6)
*x*(O)	0.30930(16)	0.30505(15)
*y*(O)	0.17406(18)	0.17617(17)
*B*_iso_(O) (Å^2^)	0.63(3)	0.10(2)
*R*_wp_ (%)	4.46	5.56
*R*_p_ (%)	2.89	3.59
*R*_p_ (%)	3.88	3.70
*R*_F_ (%)	3.90	3.57
Impurities:		
NiO (*R*–3*m*)	2.9 wt. %	4.1 wt. %
R–O × 12 (Å)	2.6110(12)	2.5879(11)
BVS(R^3+^)	3.16	3.11
Mn_SQ_–O × 4 (Å)	1.8994(12)	1.9303(12)
Mn_SQ_–O × 4 (Å)	2.7781(13)	2.7767(12)
BVS(Mn^3+^)	3.00	2.78
Ni/Mn–O × 6 (Å)	1.9711(4)	1.9572(4)
BVS(Ni^2+^/Mn^4+^)	2.91	3.02
Ni/Mn–O–Ni/Mn	137.85(7)	139.56(7)

Space group *Im*–3 (No. 204); *Z* = 2. Fractional coordinates: R: 2*a* (0, 0, 0), Mn_SQ_: 6*b* (0, 0.5, 0.5), Ni/Mn: 8*c* (0.25, 0.25, 0.25), and O: 24*g* (*x*, *y*, 0). Occupation factors, *g*, of the R, Mn_SQ_, O sites are 1. The occupation of the Ni/Mn site was fixed at 0.5Ni + 0.5Mn. For BVS of Ni^2+^/Mn^4+^, an average *R*_0_ value between *R*_0_(Ni^2+^) = 1.654 and *R*_0_(Mn^4+^) = 1.753 was used [47].

**Table 5 molecules-29-05488-t005:** Temperatures of magnetic anomalies and parameters of the Curie–Weiss fits and *M* versus *H* curves at *T* = 5 K for RMn_3_Ni_2_Mn_2_O_12_.

R	*T*_N_ (K)	*μ*_eff_ (*μ*_B_/f.u.)	*μ*_calc_ (*μ*_B_/f.u.)	*θ* (K)	*M*_S_ (*μ*_B_/f.u.)
Nd (*Pn*–3)	26	11.08(2)	11.413	−1.7(1.0)	6.90
Nd (*Im*–3)	26	11.211(13)	11.413	−0.9(7)	7.04
Sm (*Pn*–3)	23	10.704(9)	10.966	−0.2(5)	4.70
Sm (*Im*–3)	23	10.842(9)	10.966	−2.7(5)	4.70
Gd (*Pn*–3)	22	13.383(7)	13.491	+8.4(3)	10.70
Dy (*Pn*–3)	10, 36	14.886(17)	15.178	+2.8(6)	11.36

The Curie–Weiss fits are performed between 200 and 350 K using the FCC *χ* ^−1^ versus *T* data at 10 kOe. *M*_S_ is the magnetization value at *T* = 5 K and *H* = 70 kOe. *μ*_calc_ is calculated using 3.5*μ*_B_ for Nd^3+^, 1.5*μ*_B_ for Sm^3+^, 8.0*μ*_B_ for Gd^3+^, 10.6*μ*_B_ for Dy^3+^, 4.899*μ*_B_ for Mn^3+^, 2.828*μ*_B_ for Ni^2+^, and 3.873*μ*_B_ for Mn^4+^. *T*_N_ values were determined from peaks on the 10 kOe FCC *d*(*χT*)*/dT* versus *T* or *dχ/dT* versus *T* curves.

## Data Availability

The raw data supporting the conclusions of this article will be made available by the author upon request.

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
