# Peer review of "B-Site-Ordered and Disordered Structures in A-Site-Ordered Quadruple Perovskites RMn3Ni2Mn2O12 with R = Nd, Sm, Gd, and Dy"

_molecules, 2024, doi:10.3390/molecules29235488_

Round 1
Reviewer 1 Report
Comments and Suggestions for Authors
The paper brings a valuable study about RMn3Ni2Mn2O12 perovskites, focusing on their obtaining, structural characterization, and properties. The paper can be accepted after minor revision taking into account the following:
- adding morphological characterization for the relevant samples
- emphasizing the practical applications of the investigated materials.
Considering the broad audience of “Molecules”, these new data would be interesting
Author Response
Reviewer 1.
- adding morphological characterization for the relevant samples
Our reply.
We thank the reviewer for this suggestion. In the revised manuscript, we reported a scanning electron microscopy (SEM) image for one sample as an example. Very similar images were obtained for other samples.
- emphasizing the practical applications of the investigated materials
Our reply.
In the revised introduction, we added to following related to practical applications of such compounds “In addition, such perovskites can show good catalytic properties [4] as they contain transition metals in different oxidation states and exotic magnetic ground states [28].”
Reviewer 2 Report
Comments and Suggestions for Authors
This study mainly investigated the magnetic properties of different quadruple perovskites RMn₃Ni₂Mn₂O₁₂ (R = Nd, Sm, Gd, Dy), unraveling the effect of structural ordering/disordering and of the selected R on such magnetic properties. I think that the manuscript is rather interesting and almost acceptable for publication in its present form; there are just several minor issues to be addressed in the form of a minor revision:
- please add a table summarizing all your samples compositions and synthesis conditions at the end of introduction;
- I suggest you to move the materials and methods section prior results and discussion section;
- could the authors delve deep in formulating hypothesis to justify the observed Curie-Weiss temperature behavior, possibly identifying different classes of materials exhibiting it?
- conclusions should be expanded to include possible technological application of the analyzed quadruple perovskites
Author Response
Reviewer 2.
- please add a table summarizing all your samples compositions and synthesis conditions at the end of introduction
Our reply.
As the reviewer suggested, in the revised manuscript, we added a new Table 1, which summarizes all samples, their compositions and synthesis conditions.
- I suggest you to move the materials and methods section prior results and discussion section
Our reply.
We followed the journal’ style (as in the template file), where the materials and methods sections are placed in the end. Therefore, we did not change the order.
- could the authors delve deep in formulating hypothesis to justify the observed Curie-Weiss temperature behavior, possibly identifying different classes of materials exhibiting it?
Our reply.
Unfortunately, at the moment, we do not know other examples (classes of materials) with nearly zero Curie-Weiss temperature (of course, with long-range magnetic orders, not paramagnetic samples). Other discussions of the observed Curie-Weiss temperature behavior are given as below
“It is interesting that the Curie–Weiss temperatures (q) were very small suggesting that antiferromagnetic and ferromagnetic interactions are of the same magnitude and nearly cancel each other. In GdMn3Ni2Mn2O12 and DyMn3Ni2Mn2O12 compounds, the Curie–Weiss temperatures were slightly positive suggesting that ferromagnetic interactions are slightly stronger….. In all R2NiMnO6 (R = La-Lu [10, 16-19]), Tl2NiMnO6 [23], and In2NiMnO6 [20], positive and large Curie–Weiss temperatures were found indicating that ferromagnetic interactions are dominant. On the other hand, in Sc2NiMnO6 [22], a negative Curie–Weiss temperature of about -60 K was observed indicating that antiferromagnetic interactions are dominant. Therefore, there could be strong competition of different magnetic interactions in RMn3Ni2Mn2O12 resulting in nearly zero Curie–Weiss temperatures.”
- conclusions should be expanded to include possible technological application of the analyzed quadruple perovskites
Our reply
In the revised introduction, we added to following related to practical applications of such compounds “In addition, such perovskites can show good catalytic properties [4] as they contain transition metals in different oxidation states and exotic magnetic ground states [28].” We cannot state practical applications in the conclusion section as we did not perform such tests in our paper.
Reviewer 3 Report
Comments and Suggestions for Authors
This article investigates the synthesis, structural properties, and magnetic behaviors of A-site-ordered quadruple perovskites with different rare-earth elements (Nd, Sm, Gd, and Dy). The authors explore the effects of B-site cation ordering and report two structural modifications that arise from varying synthesis temperatures: Pn-3 with B-site ordering and Im-3 with B-site disorder. Their work suggests that, unlike R2NiMnO6 perovskites, the magnetic properties are relatively unaffected by the ordering state at the B site. The study contributes to understanding the magnetic behavior of A-site-ordered quadruple perovskites, adding valuable insights into how synthesis conditions affect structural and magnetic properties. The authors employ synchrotron X-ray diffraction and SQUID magnetometry, ensuring precise measurements for structural analysis and magnetic behavior. Figures for magnetic susceptibility, diffraction patterns, and specific heat capacity provide visual clarity and reinforce their findings. Overall, this paper makes a valuable contribution to the field of perovskite materials, the data were convinced to me and the manuscript is well written. I suggest to accept this manuscript after few minor revisions.
1. Since the accuracy is related with the crystal domain size and purity, the image of the crystal is encouraged to be given.
2. The magnetic transition temperatures are stated, but the discussion could benefit from additional comparisons to similar compounds to explain the reason of the identical results.
3. Although the Curie-Weiss temperatures are near zero, suggesting antiferromagnetic and ferromagnetic interactions are balanced, a deeper explanation of the implications of this balance would be useful.
Author Response
Reviewer 3.
- Since the accuracy is related with the crystal domain size and purity, the image of the crystal is encouraged to be given.
Our reply.
We thank the reviewer for this suggestion. In the revised manuscript, we reported a scanning electron microscopy (SEM) image for one sample as an example. Very similar images were obtained for other samples.
- The magnetic transition temperatures are stated, but the discussion could benefit from additional comparisons to similar compounds to explain the reason of the identical results.
Our reply.
In the revised manuscript, we added the following discussion “Magnetic transition temperatures and properties did not show any clear systematic trends as a function of the ionic radius of R3+ cations in RMn3Ni2Mn2O12. On the other hand, in R2NiMnO6 perovskites, magnetic transition temperatures show a clear and sharp decrease with decreasing the ionic radius of R3+ cations [18, 19, 25, 28].”
- Although the Curie-Weiss temperatures are near zero, suggesting antiferromagnetic and ferromagnetic interactions are balanced, a deeper explanation of the implications of this balance would be useful.
Our reply.
Other discussions of the observed Curie-Weiss temperature behavior are given as below
“It is interesting that the Curie–Weiss temperatures (q) were very small suggesting that antiferromagnetic and ferromagnetic interactions are of the same magnitude and nearly cancel each other. In GdMn3Ni2Mn2O12 and DyMn3Ni2Mn2O12 compounds, the Curie–Weiss temperatures were slightly positive suggesting that ferromagnetic interactions are slightly stronger….. In all R2NiMnO6 (R = La-Lu [10, 16-19]), Tl2NiMnO6 [23], and In2NiMnO6 [20], positive and large Curie–Weiss temperatures were found indicating that ferromagnetic interactions are dominant. On the other hand, in Sc2NiMnO6 [22], a negative Curie–Weiss temperature of about -60 K was observed indicating that antiferromagnetic interactions are dominant. Therefore, there could be strong competition of different magnetic interactions in RMn3Ni2Mn2O12 resulting in nearly zero Curie–Weiss temperatures.”